# Current Methods for Analysing Mesenchymal Stem Cell-Derived Extracellular Vesicles

**DOI:** 10.3390/ijms25063439

**Published:** 2024-03-19

**Authors:** Ana Kolenc, Elvira Maličev

**Affiliations:** 1Blood Transfusion Centre of Slovenia, Šlajmerjeva 6, 1000 Ljubljana, Slovenia; ana.kolenc@ztm.si; 2Biotechnical Faculty, University of Ljubljana, Jamnikarjeva ulica 101, 1000 Ljubljana, Slovenia

**Keywords:** mesenchymal stem cell-derived extracellular vesicles, extracellular vesicle characterisation, protein determination, flow cytometry, particle number determination

## Abstract

The use of extracellular vesicles (EVs) generated by mesenchymal stem cells (MSCs) holds great promise as a novel therapeutic approach. Although their immunomodulatory and regeneration potential has been reported to be similar to that of MSCs, the use of MSC-derived EVs in clinical settings will require several problems to be resolved. It is necessary to develop a standardised and widely accepted isolation technology and to improve methods such as the quantification and characterisation of MSC-derived EVs. In this way, EV studies can be compared, the acquired knowledge can be safely transferred to clinical platforms and the clinical results can be evaluated appropriately. There are many procedures for the collection and analysis of vesicles derived from different cells; however, this review provides an overview of methods for the determination of the total protein amount, specific proteins, particle number, non-protein markers like lipids and RNA, microscopy and other methods focusing on MSC-derived EVs.

## 1. Introduction

The unique ability of mesenchymal stem cells (MSCs) to regulate tissue repair and influence the immune system makes them an attractive therapeutic tool for the treatment of various diseases. While the differentiation potential of MSCs was the primary focus of their therapeutic efficacy, it eventually became apparent that their paracrine effect on a variety of cells was clinically pronounced. Since MSCs have been successfully applied in graft-versus-host disease, the number of studies using autologous or allogeneic MSCs to treat immune and inflammatory disorders is rapidly increasing. MSCs show potential to mitigate Crohn’s disease, multiple sclerosis, osteoarthritis and diabetes mellitus. Besides this, their tissue regeneration potential has been reported for different conditions, including pulmonary fibrosis, spinal cord injury, myocardial infarction, knee cartilage injury and some neurodegenerative diseases.

Despite the observed clinical effects of MSCs, their low survival rate and retention in the body after application have been noticed [1]. It is also mentioned that MSCs may accumulate in the lungs following intravenous infusion [2]. These findings may delay their more extensive clinical implementation. Since the reports of many studies show that MSCs mainly exert their therapeutic effects through the secretion of active molecules, MSC-derived extracellular vesicles (EVs) have gained attention [3]. Compared to cells, EVs are supposed to have similar therapeutic effects to MSCs but are safer to apply due to their cell-free properties. Extracellular vesicles are significantly smaller than cells and range in size from 50 nm to over 1 um. They belong to the heterogeneous group of membrane structures classified into two distinct classes, exosomes and microvesicles, both carrying bioactive molecules. The primary form of EV is a bilayer consisting of phospholipids and cholesterol, sphingomyelin and proteins, enclosing a lumen rich in bioactive cargo. Cytokines, growth factors, signalling lipids, mRNAs and regulatory miRNAs are responsible for most paracrine functions of MSCs [4]. Therefore, EV-based treatment could be a promising therapeutic substitute for cell therapies. Due to their small size, EVs can bypass most physiological barriers and reach their target tissues [5]. EVs can easily pass across the blood–brain border, making them especially interesting for neurodegenerative diseases. EVs have another advantage over the application of cells since EVs can be utilised in clinical settings by filtration-based sterilisation.

Early proteome analyses of EVs produced from MSCs have shed light on the functional proteins within them and, in turn, on possible therapeutic uses [6,7,8]. The therapeutic potential of MSC-derived EVs has been reported to be similar to that of MSCs, both in immune regulation and tissue regeneration. EVs affect several target cells, including macrophages, microglia, chondrocytes, articular chondrocytes, endothelial cells, fibroblasts, pericytes, neural stem cells, neurons, hepatic stellate cells and podocytes. Some clinical trials that have used MSC-derived EVs have already been registered on the Clinical Trials Database (https://ClinicalTrials.gov/, accessed on 13 March 2024). As of March 2024, 18 clinical trials are included in the database, of which 13 are ongoing and five have been withdrawn or completed. Currently, most clinical studies focus on respiratory diseases (acute respiratory distress syndrome, COVID-19, bronchopulmonary dysplasia), as well as wound healing (burns) and other diseases (ulcerative colitis, liver failure syndrome, Crohn’s disease and retinitis pigmentosa). The effects of EVs are promising; however, ensuring the integrity of vesicles and maintaining their biological activity and reproducibility to ensure homogeneity in the final product remain challenging [9]. The lack of a method to scale up the production of EVs with customised therapeutic characteristics is a significant clinical barrier to the use of MSC-EVs. To manufacture clinical-quality EVs, robust, affordable and scalable methods for EV production and separation should be implemented together with more accurate analytical techniques, like counting, EV size-based methods, phenotyping and other methods.

The first minimal requirements for EVs were published in 2014, co-authored by the International Society for Extracellular Vesicles (ISEV) Board and now referred to as the Minimal Information for Studies of Extracellular Vesicles (MISEV2014) [10]. This was followed by a lengthier, more comprehensive document in 2018 and recently in 2024 [11,12]. The MISEV2023 guidelines have been drawn up with the help of 1051 co-authors from 53 countries. The EV-TRACK program also offers an open-access platform for the recording of crucial data for EV publications [11,12]. However, these guidelines are for EVs in general, not specifically for MSC-derived EVs. Many methods have been developed and implemented to allow MSC-derived EV characterisation, but none of these techniques enable their complete characterisation. This review highlights the methods currently used to analyse MSC-derived EVs (Figure 1). These methods allow total and specific protein amount determination, particle number and size determination, the detection of non-protein markers like lipids and RNA, microscopy and others. Besides briefly describing the protocols, we summarise the experience and findings of different groups analysing MSC-derived EVs applied in science and clinical applications.

## 2. Considerations before MSC-EV Analysis

One of the most important factors to consider when assessing MSC-EVs is the source of the vesicle-releasing cells (Figure 2). Further, EVs are collected from cell cultures with MSCs, and the procedures prior to collection can affect the EVs. The cell viability, the passage number, the culture medium’s composition, the culture conditions (temperature, pH, CO2 concentration, culture duration) and the process of isolation are some parameters that affect EVs’ yields, composition and function. MSC-EVs have been isolated and concentrated using various methods (precipitation, chromatography, filtration, immune precipitation, ultracentrifugation, etc.); however, most of them do not allow the isolation of the pure population of EVs. At present, reporting the steps taken during the pre-processing and actual sample collection of EV-rich material is crucial in order to enable the comparison of data obtained from the following analyses. According to the most recent guidelines, the focus should be on providing as much information as possible regarding the EVs’ collection, sorting, concentration, storage and characterisation for further research [12].

## 3. Protein Determination

### 3.1. Total Protein Amount Detection

The determination of the particle number or total protein amount is the most commonly used approach to assessing the number of EVs, using various methods, including colourimetric and fluorimetric assays. The Bicinchonic Acid (BCA) Protein Assay is a copper-based assay for the colourimetric detection and quantification of total protein. The base is the reduction of copper in the presence of peptides containing three or more amino acid residues. The added BCA reagent reacts with the cuprous cation, producing a water-soluble BCA–copper complex that exhibits absorbance at 562 nm. The total protein concentration can be determined by applying a linear fit (20–2000 g/mL) to the standard curve of common proteins [13]. Micro BCA™ Protein Assay Kits (1–40 g/mL) are also used to cover the wide range of protein content in the isolated EVs. Various other commercial kits are available: one is the fluorometric NanoOrange Protein Quantitation Kit, which the manufacturer claims has less protein-to-protein variability [14].

In recent MSC-EV studies, the BCA Protein Assay Kit was used to quantify samples of EVs isolated from umbilical cord matrix MSCs (UCM-MSCs) [15], bone marrow MSCs (BM-MSCs) [15,16] or adipose stem cell (ASC) cultures [15,17]. MSCs were cultured (passage 4 or 5) before actual EV-rich samples were collected [15]. The MISEV2023 guidelines advise against using the detected protein concentration as an equivalent to the EV concentration since different cellular phenotypes or stimulations may cause the release of specific proteins. Generally, the concentration is determined by using the bovine serum albumin standards [15,16,17]. However, protein quantification can be overestimated due to co-isolated protein contaminants [11,12]. Franquesa et al. (2014) concluded that protein measurements of ASC-EV samples are insufficient to quantify EVs since the pellets of EVs obtained by high-speed ultracentrifugation contain protein aggregates, lipoprotein particles and other impurities [18]. The MISEV2018 guidelines suggest simultaneously using several quantification methods for EVs in order to overcome the issues related to protein determination. It is also important to define whether a detergent is used to break down the EVs in the study. In this instance, both membrane and intracellular proteins are quantified [11]. Additionally, the BCA assay can also be influenced by the fluorescent dyes (such as PKH67) used in studies to determine the ability of MSC-EVs to act as potential drug carriers [19].

### 3.2. Specific Protein-Based EV Detection

Western blotting and flow cytometry are the most used methods to determine the specific proteins in EV samples. To prove the presence of EVs in samples, three categories of EV markers that must be analysed have been proposed [11] (Figure 3). At present, we do not have a standard marker that would enable the identification and characterisation of all extracellular vesicles in samples, because none of them are specifically expressed on MSC-EVs (both exosomes and microvesicles) [18,20]. Therefore, it is necessary to analyse at least one protein from each of the following categories: (1) transmembrane or GPI-anchored proteins localised at the external membranes of cells or MVs; (2) cytosolic proteins or periplasmic proteins; and (3) major components of non-EV structures often co-isolated with EVs. Proteins in the first category present the unique lipid bilayer structures of EVs. Proteins in the second category determine the intracellular material within the same structure, and proteins in the third category confirm the EV sample’s purity level. Two additional categories enable the evaluation of proteins specific to the identification of small EV subtypes [11].

#### 3.2.1. Western Blotting

Western blotting is a process that involves size-based protein separation on polyacrylamide gel electrophoresis, protein transfer and membrane immobilisation. To evaluate whether the investigated proteins are enriched in EVs instead of their producing cells, the process should be applied by loading EV samples and cell culture material lysates side-by-side (previous protein amount determination is needed) [11,12]. Since they are typically found on EVs from a range of cell types, the transmembrane proteins CD9, CD63 and CD81—collectively known as tetraspanins—are determined most of the time [21]. In addition to these proteins, the expression of CD90, CD73 [22], CD105 [23], programmed death-ligand 1 protein (PD-L1) [24], glyceraldehyde 3-phosphate dehydrogenase (GAPDH) [15], syntenin-1, calnexin [15,23,25,26], glial fibrillary acidic protein (GFAP) [25], apoptosis-linked-gene-2-interacting protein X (ALIX) [26,27], tumour susceptibility gene 101 (TSG101) [25,26,27], amyloid-beta (Aβ), cis-Golgi matrix protein (GM130) [28] and heat shock protein 70 (HSP70) [29] is measured in research on MSC-derived EVs. The above analysis used purchased cultures of human BM [25,28] and human placenta-derived MSCs [27], MSCs isolated directly from human BM tissue [22,23], mouse BM [24], chorionic villus tissue from a human placenta [26] and human tonsil tissue [29]. Samples are usually run on a 4–20% polyacrylamide gel before being deposited onto a nitrocellulose membrane and treated with the primary and secondary antibodies conjugated with horseradish peroxidase (HRP) [15,25,26]. Antigen-positive and negative control samples, along with purity controls, should be included in the analysis.

#### 3.2.2. Flow Cytometry

Some flow cytometers can detect fluorescently labelled EVs; therefore, the concentration, phenotype, refractive index (RI) and size of EVs can all be determined. A greater fluorescence intensity signal is produced when an EV has more fluorophores bound to it. Since the result is a signal for a specific protein marker in the overall EV population, these techniques are population-level rather than single-EV procedures [11]. However, Shen et al. (2018) described a single-EV flow cytometry analysis technique based on the target-initiated engineering of DNA nanostructures on each EV to find statistically significant variations in the molecular signatures and distinguish those originating from cancer cells in a heterogeneous sample [30]. Flow cytometry can characterise particles with a high count rate, offer statistically significant information to distinguish populations of EVs and measure the concentration and particle diameter [31]. However, most commercial flow cytometers were developed to examine cells and cannot be simply adapted for EV measurement. To compare data amongst flow cytometers, the limit of detection (LoD; the lowest signal level that can be differentiated from background noise) and the optical setup of a flow cytometer must, therefore, be standardised [31]. Various scientists have adopted various strategies to discriminate MSC-EVs from debris before detecting actual EV markers: lactadherin–FITC was used to detect phosphatidylserine (PS) in an MSC-EV bilayer, although the majority of the exosomes remained undetected [22]; CD63-coated beads were used to capture vesicles with a magnet [32]; and CFDA–SE was used to identify intact EVs [17]. The latter was performed on ASC-derived EVs, and, in addition to using CFDA–SE to distinguish between debris, membrane fragments and EVs, they used fluorescent beads for the easier determination of the particle size prior to staining. Generally, beads are also used to capture either all particles in the sample, regardless of the composition, or specific particles based on bead-conjugated antibodies [12]. EVs are then usually stained for MSC surface markers CD73, CD90 [22,32] and CD105 [33,34], along with tetraspanin markers CD9, CD63, and CD81 binding to different fluorescent dyes [17,22,32,33]. The presence of hematopoietic markers CD45, CD44 and CD29, the α4 and α5 integrins and human leukocyte antigen (HLA) is also analysed [35]. Additionally, it is crucial to use particles to calibrate and gate a trigger signal in the proper proportion, control the flow rate and record the events. By treating the samples with 0.25% TritonX-100 to break up the vesicles, lyse control was included to demonstrate that the signals in the EV fraction were truly dependent on the presence of intact EVs [22,32]. In one study, instead of staining the isolated EVs, the authors used PKH-67 fluorescent dye to label the actual suspension of ASCs before culturing. However, they could not separate the EV fluorescence from noise [18]. Commercial bead-based multiplex approaches have also been developed and are frequently used for surface marker characterisation when using flow cytometry. Isolated EV samples are exposed to capture beads for up to 37 surface markers and a CD9, CD63 and CD81 detection antibody cocktail. In this specific instance, the beads with bound EVs are evaluated on a MACSQuant Analyzer, where the median fluorescence intensity for each capture bead and the signal strength for each bead population are measured [36].

One of flow cytometry’s newest and growing applications is imaging flow cytometry, which allows single-EV analyses with enhanced fluorescence sensitivity, a minimal background and the capacity for image confirmation. The labelling of CD9, CD63 [23,37] and CD81 [23] was used in a study of BM-MSC-EVs as drug delivery systems [37] and as a potential therapy for rheumatoid arthritis [23]. Fluorescent polystyrene beads are a very helpful tool in accurately acquiring EVs and identifying the area on the dot plot where the EVs should be. Many controls are also often used, including a buffer only, single stain control, isotype control, detergent control and unstained control [38]. Furthermore, as EVs are typically examined immediately after staining, without washing off the unbound antibodies, some (regardless of the EV source culture) emphasise the need for antibody titration [39]. Inappropriate antibody concentrations might lead to excessive background noise and non-specific binding [40].

Intriguingly, a study of hair follicle-derived MSCs notes that the outcomes of flow cytometry and Western blot analyses are different [36]. The causes of this may be related to the technical distinctions between the two methods; Western blotting reveals the contribution of all proteins in the preparation, whereas bead capture only assesses exposed epitopes. In summary, flow cytometry can be challenging because of the heterogeneity of EVs, particularly in size, but it also offers quantitative data, the high-throughput simultaneous analysis of multiple parameters and the ability to identify individual EVs. It also requires specialised technology and standards. In contrast, Western blotting offers only semi-quantitative data on a single protein at a time, although it is helpful in verifying the existence of particular proteins in EVs or on their surfaces. Therefore, the decision regarding which of them to use will depend on the goals and research context.

Combining several approaches is frequently advised in order to develop a thorough understanding of EVs [11].

#### 3.2.3. Mass Spectrometry

Mass spectrometry (MS) enables high-throughput peptide profiling [41] and, consequently, the detection and characterisation of EV-associated proteins [12]. Since there have been reports linking EVs with the progression of some diseases, this technique can also be used to explore the functions of EVs and identify EV proteins as potential biomarkers (for prognosis and diagnosis) [41]. We differentiate between untargeted analysis, which is used to find every detectable ion in the sample, and targeted analysis, which is most helpful in characterising EV purity [12].

Several early studies used trypsin-digested EV isolates for direct liquid chromatography–mass spectrometry (LC–MS) EV analyses. The results were used to support flow cytometry or Western blotting data. However, recent studies have been successful in obtaining the coverage of EV proteomes with ultra-performance LC (UPLC), using the fractionation of the EVs’ protein components (with techniques like SDS-PAGE) to reduce the interference of lipids and high-abundance proteins in the analysis. Still, it may also reduce the analysis’s sensitivity [42]. LC–MS was utilised by Peltzer et al. (2020) to analyse the presence of BM-MSC-EV-associated proteins [33]. They had previously undergone 10% SDS-PAGE separation, direct reduction, alkylation and trypsin digestion in gel fragments. Proteins were identified and quantified by comparing the results with a database of human proteins, where most of the proteins analysed were already annotated as exosomal [33]. The proteome of the same source culture type was also investigated with quantitative MS. They determined whether cell growth conditions in 2D or 3D affected protein expression. When comparing EVs to donor cells, they found that whilst markers for cytosol, nuclei and other organelles were minimally expressed, extracellular markers, microparticles, plasma membranes and other cellular components were enriched in EVs [43].

#### 3.2.4. Enzyme-Linked Immunosorbent Assay

An enzyme-linked immunosorbent assay (ELISA) can also be used as a method of EV quantification based on one or more specific proteins or other molecules. Direct, indirect, competitive and sandwich approaches are the four methods that can be applied [44]. Commercially available ELISA kits are usually used. Purified EV preparations or EV lysates may be placed directly on a solid support that has been prepared with an immobilised capturing antibody. The captured vesicular targets are then exposed to a secondary detection antibody [42]. In a study on EVs isolated from ASCs, the researchers immobilised EV samples directly onto the wells of microtiter plates. They added detection antibodies binding to tetraspanins CD63, CD9 and CD81 and a secondary antibody linked to the HRP enzyme. A microtiter plate reader quantified the results after adding a colourimetric substrate [18]. It is important to note that ExoELISA kits are in use to execute ELISA on EV samples, but each focuses on detecting only one protein. If samples for all three significant tetraspanins are to be analysed with ELISA, three commercial kits should be used for each sample. A human MSC experiment used ELISA to identify the signalling molecules that hMSCs release and that can elute with EVs and affect signalling. In order to analyse the concentrations of these components, they chose vascular endothelial growth factor (VEGF) as a typical indicator of their presence [14]. Franquesa et al. (2014) argue that since we can only analyse vesicles with the particular antigens for which we have antibodies, it is impossible to quantify all EVs in our sample in this way [18]. This is due to the concerns already discussed at the beginning of this section.

## 4. Particle Number Determination

### 4.1. Nanoparticle Tracking Analysis

Nanoparticle tracking analysis (NTA) can be used to measure the absolute size distribution and concentration. The camera records each particle’s path (moving in a Brownian motion) after a light beam strikes it, causing the particles to scatter light. The size distribution is based on a hydrodynamic diameter through the Stokes–Einstein equation, and the number of particles in the field of view is determined [41]. Multiple technical readings are obtained on an instrument equipped with a 405 nm [15,17,45,46] or 488 nm laser [25]. While Las Heras et al. (2022), working on hair follicle-derived MSC-EVs, obtained medium–large EVs (up to 700 nm) [36], MSC-EV samples analysed with NTA usually show a size distribution profile mainly enriched in small EVs (<200 nm) [15,26,27]. According to the above-cited articles, the size of the vesicles could be determined by the cells of origin. We believe this to be the case because they used five distinct MSC types while still applying the same isolation method.

Moreover, accurate quantification might only be achievable within a specific concentration and size range that depends on the analyser and execution [11]. Since light scattering techniques are not EV-specific and record co-isolated particles such as lipoproteins and protein aggregates/complexes, they lead to overestimated EV numbers [11,12,18]. Thus, the MISEV2023 guidelines recommend using buffer-only control and reporting the instrument settings along with the final diameter distributions, rather than single-size statistics [12]. According to Witwer et al. (2019), in their guideline for the definition of MSC-derived EVs, the ratio of membrane lipids to protein or RNA should be a definite, more quantifiable characteristic than the results produced by particle analysis [47]. Almeria et al.’s (2019) observation of MSC-EV concentrations that were more than three orders of magnitude greater when using NTA instead of flow cytometry is an example of insufficient method reliability [22]. One explanation might be that the MSC-EVs were not isolated in advance and that the analysis was performed using a direct supernatant/conditioned MSC culture medium [22].

### 4.2. Dynamic Light Scattering

By correlating the variations in the scattered light intensity brought about by the Brownian motion of the particles, dynamic light scattering (DLS) can also determine the size of the particles. Unlike NTA, which tracks individual particle scattering, DLS measures bulk scattering, i.e., changes in scattering intensity from a bulk sample [41]. In other words, NTA produces a number-based distribution, and DLS produces an intensity-based distribution [48]. For studies of MSC-EVs, samples are diluted in DPBS [24] or 0.15 M NaCl [28] and placed into a quartz cuvette to analyse. An instrument equipped with a 532 nm laser is needed [28,29]. An analysis of murine BM [24] and human UCM [49] MSC-EVs resulted in an average diameter of around 100 to 200 nm. The same was discovered in human placenta-derived MSC-EV samples [50]. However, Welsh et al. (2024) now point out that DLS should only be used to validate the presence of submicrometer particles; it should not be employed quantitatively unless a monodisperse size fraction of EVs is used [12].

### 4.3. Resistive Pulse Sensing

In some studies on MSC-EVs, resistive pulse sensing (RPS) or tunable RPS (TRPS) is used to determine the particle concentration and size distribution. Unlike NTA and DLS, RPS is a non-optical technique [12]. This method divides two fluid chambers—one carrying the sample and the other an electrolyte solution—by a membrane with tunable pores. A voltage is applied across a membrane as the particles move through. This causes a pulse that is precisely proportional to the volume of the particles, whereas the blockade rate is related to the particle concentration [51]. TRPS was used to analyse ASCs and BM-derived MSC-EVs. The authors acquired the size of 300–400 nm for medium EVs and 100–150 nm for small EVs [52]. Symonds et al. obtained comparable findings on ASC-EVs, but they used size exclusion chromatography in addition to ultracentrifugation for particle isolation [53]. These methods may be biased towards certain particle size ranges because of the pore sizes, sensitivity, etc., and the proprietary software used for data analysis.

In conclusion, for particle number determination, NTA and DLS are complementary methods: NTA provides a number-weighted size distribution and a relative particle concentration, whereas DLS provides a rapid assessment of polydispersity and an intensity-weighted particle distribution. RPS is limited in its pore diameters and seems to be better suited to the identification of larger vesicles, although it still offers information on the particle size and concentration. When studying MSCs, the choice ultimately relates to the properties of the studied samples and the resources that are accessible, even if NTA and DLS are used more frequently.

## 5. Non-Protein Marker Detection

### 5.1. Transcriptomics

Both DNA and RNA have been detected in EVs and are considered significant contributors to the molecular events occurring in the recipient cell [17]. Dyes can detect RNA in EV preparations, although some dyes may also detect non-EV-associated RNA [11]. Thus, the EVs’ total RNA is usually isolated and treated with DNase. cDNA libraries are generated using adaptor-specific primers and sequenced on tools like the Illumina Novaseq. Gene ontology enrichment analysis, miRNA target prediction and miRNA identification using mRNA search bases are carried out [54]. Libraries of EV samples are highly enriched in the classes of rRNAs, tRNAs, mRNAs, miRNAs and piRNAs [17]. Sequencing has revealed the presence of several miRNAs involved in MSC differentiation, migration or immune-modulatory functions, cancer growth inhibition and cardioprotection [46]. The total RNA of EV samples is usually quantified by RT-PCR [16] or real-time qPCR [17]. However, Peltzer et al. (2020), researching BM-MSC-EVs, state that because qPCR panels are based on probes or primers, they enable a thorough identification of the miRNA [33]. Consequently, more miRNAs might be discovered by RNA-Seq research. The miRNA expression within MSC-EVs could also vary according to the tissue and/or species [33]. RNA purification and pre-assay preparation are once again emphasised, regardless of the actual characterisation technique [12].

### 5.2. Lipidomics

According to MISEV2018, phospholipids found in EV lipid bilayers can determine the presence of EVs in a sample. Extensive EV lipidomic profiling studies are required to better understand EVs’ formation and protein packing mechanisms and specify new EV molecular markers [42]. PS can be indirectly detected by binding fluorescently labelled proteins such as Annexin V or the C1C2 domain of lactadherin/MFGE8, although these molecules are not exclusive to EVs [11]. Another technique begins with liquid–liquid phase extraction and the isolation of lipids and proceeds with a high-resolution mass spectrometer, gas chromatography–MS, LC–MS or direct infusion electrospray ionisation [42].

To date, lipidomic studies have been utilised while working with plasma and urine-derived extracellular vesicles. Authors have used LC [55] or UPLC [56] in combination with tandem mass spectrometry. The UPLC system was recently used to profile the primary phospholipids and sphingolipids in murine MSC-EV samples. Ceramides, sphingomyelins, phosphatidylcholines, phosphatidylethanolamines, PS, phosphatidylinositols, eicosanoids and cholesterol were found after the lipidomics analysis. They have all been previously reported to be enriched in exosomes [46].

## 6. Other Characterisation Methods for Extracellular Vesicles

### 6.1. Electron Microscopy

Techniques that produce high-resolution images of single EVs, including atomic-force microscopy (AFM) and electron microscopy (EM), are irreplaceable regarding the data that they produce. EM is widely used to demonstrate the existence of EVs detected by other methods (e.g., flow cytometry) by showing photos of the actual EVs in contrast to non-EV particles [31]. We can evaluate the morphology and size, but it is challenging to actually quantify EVs [18]. MSC-EV samples are usually loaded onto nitrocellulose [45] or formvar grids coated with carbon [23,28] or copper [24]. Precisely due to this film on the grid, we are only left with EVs larger than the openings on the film, causing inaccurate counting [57]. Samples are then fixed in paraformaldehyde with glutaraldehyde and contrasted in 2% [18,43,45,56] or 1% [24] uranyl acetate. Cone et al. (2021) stained samples with anti-CD63 antibodies and gold-labelled secondary antibodies before contrasting with uranyl acetate, while some have used 2% phosphotungstic acid [25,50,54].

MSC-EV samples from different tissues are usually examined under a scanning (SEM) or transmission electron microscope (TEM) [18]. Cryogenic TEM (cryo-TEM) allows for the characterisation of EVs close to their natural structures because samples are immediately placed into an EM grid, vitrified and observed [58]. Samples for cryo-EM are fixed by plunge freezing at −180 °C in liquid ethane [36,46] or at −170 °C in liquid nitrogen [14] before being observed with transmission EM operation [14,36,46]. In contrast to cryo-EM, methods such as NTA and DLS that depend on particle mobility are more likely to provide an overestimation of the EV size. However, once more, the challenge is in assessing the complete population of EVs in the sample, as high-resolution imaging has a relatively low throughput and many EVs may still not be observed [12].

A study comparing the size distribution of EVs with NTA and EM revealed comparable results; however, microscopy offered greater small EV counts, in addition to the ability to identify non-EV particles based on their size and morphological characteristics. They used SEM to examine lung cancer cell line EVs [59]. Therefore, microscopy produces unique and very important data for EV characterisation. Nevertheless, when using EM for MSC-EV characterisation, it is suggested that cryo-TEM should rather be used because of its maintenance of the original EV morphology [12].

### 6.2. Atomic Force Microscopy

AFM, which can provide diverse data on the three-dimensional topography, size and other biophysical features of EVs, is a widely accessible and affordable alternative to cryo-TEM. Vesicles must first be immobilised on a substrate via electrostatic fixing on a positively charged substrate, anchoring to a functionalised surface or trapping in a filter before imaging and image analysis can be performed [60]. AFM has already been used to identify the presence of EVs in isolates from a conditioned medium of UCM-MSCs, BM-MSCs and ASCs, where both individual vesicles and vesicle aggregates of various sizes were seen [15]. Image analysis is then performed on software like ImageJ (https://imagej.net/ij/) [61]. It was discovered that the vesicle sizing determined by the AFM data and cryo-TEM imaging is compatible, although it is crucial to minimise sample evaporation while working with hydrated vesicles. We do not wish to increase the surface concentration or produce surface deposition artefacts [60].

### 6.3. Zeta Potential

Zeta potential (ZP) is used to determine the surface potential of EVs and can specify the stability of particle–particle and particle–medium interactions, including the tendency of the particles to aggregate. ZP is, therefore, one of the best instruments in examining the EV’s role in biological processes regarding cellular uptake and cytotoxicity. It is also used as an indicator of the surface charge and colloidal stability [62]. For zeta potential analysis, EV samples are loaded into disposable zeta cells with gold electrodes. An analysis of UCM-MSCs, BM-MSCs and ASCs was performed on the Zetasizer Nano ZS [15,28]. The zeta potential was also assessed via DLS in a study on human placenta-derived MSC-EVs [50] and murine MSC-EVs [46] and via NTA on rat BM-MSC-EVs [63]. De Almeida Fuzeta et al. (2022) discovered that three separate tissues of MSC-EVs retained the EV surface charge relatively consistently (between −15 and −19 mV) [15]. However, the results of other studies vary slightly: while some found a peak in the zeta potential at about −26.3 mV [28], others observed that the mean of the zeta potential in rat BM-MSC-EVs was −42.06 mV [63] and −16.3 mV in ASC-EVs [64].

## 7. Summarised Recommendations

The updated MISEV2023 guidelines offer new terminology relating to EVs, discuss pre-analytical variables and EV sources, describe how to isolate EVs and offer methods for the recognition and analysis of EVs’ properties. However, the most crucial are the suggestions regarding the particular reporting guidelines for various EV characterisation techniques (Table 1) [12].

EVs can be produced by various cell types and are mainly studied as biomarkers and drug delivery systems. EV analysis methods and guidelines are described for EVs in general, but it is crucial to specify MSC-EVs’ unique characteristics, as they have the greatest potential in therapeutic applications. While MSC-EVs and other EVs share common methodologies for isolation, their cargo composition, biological functions and therapeutic applications distinguish them. Even within the tissues from which they originate, MSC-EVs show variations, which may contribute to their therapeutic efficacy and further emphasise the necessity for technique specificity. As already mentioned, at least two approaches are nearly always combined when analysing MSC-EVs. Particle size and concentration measurements are the most widely used techniques. We suggest the standard criterion for MSC-EVs: particles released from MSCs cultured in vitro with the expression of CD9, CD63, CD81, CD105, CD73 and CD90. These are supplemented by the evaluation of specific cargo proteins or surface markers to verify that the outcomes of the particle tracking analyses are, in fact, connected to vesicles and not to non-vesicular particles. To support this, morphological investigations, which often employ electron microscopy to provide tangible image material as proof of the existence of EVs, are used [37,65,66].

In the future MSC-EV studies, we recommend reporting all crucial information about the used methods (from sample collection to isolation, detection and characterisation) and instruments, as well as using multiple synergistic methods. In addition to the biophysical properties, we also encourage a focus on the functional aspects and assessment of EV-mediated effects.

## 8. Conclusions

The use of MSC-derived EVs for therapy, as an alternative to MSCs, confers several advantages. However, standardised procedures for MSC-derived EV collection and characterisation are necessary to allow data to be compared. This will help to advance the field and adequately transfer preclinical research to clinical platforms. Researchers have used different approaches that are suitable for the counting and determination of MSC-EVs, and we have described the most promising of them in this review. We found that EV isolation with as little contamination as possible presents a great challenge. As a result, precise EV quantification is difficult. Determining the significance of particular EV populations and connecting them to pathogenic pathways and therapeutic approaches is still challenging. Specific MSC-EV characterisation techniques are required to fully understand their therapeutic potential, guarantee their safe use in clinical settings and distinguish between various types of MSC-EVs for specialised treatments. In summary, to tackle these issues, we call for interdisciplinary cooperation, technical development and thorough technique validation.

## Figures and Tables

**Figure 1 ijms-25-03439-f001:**
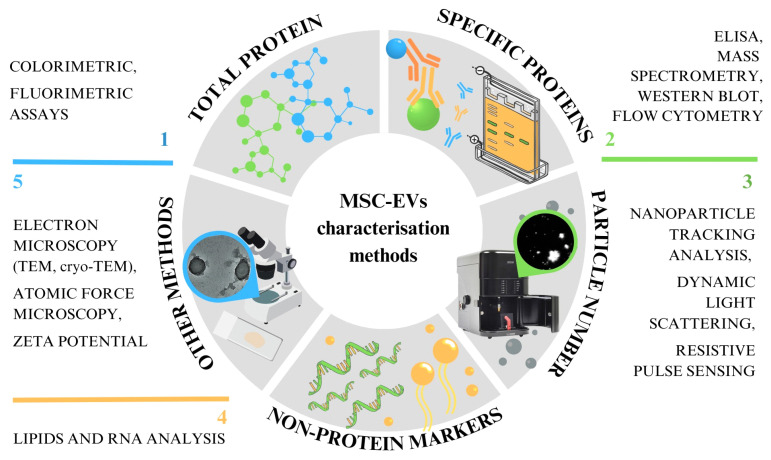
Abstract of different methods of characterising EVs, which can be roughly divided into five categories.

**Figure 2 ijms-25-03439-f002:**
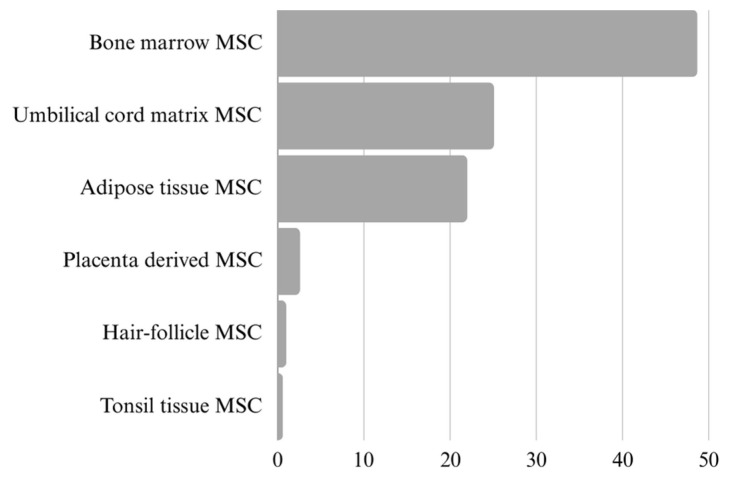
Ratio presentation of different MSC sources in research publications on extracellular vesicles (N = 626). The data were obtained from the Web of Science, refined by publication year (2019–2024), document type (article) and specific cell source. It is important to recognise that many publications include multiple cell sources at once.

**Figure 3 ijms-25-03439-f003:**
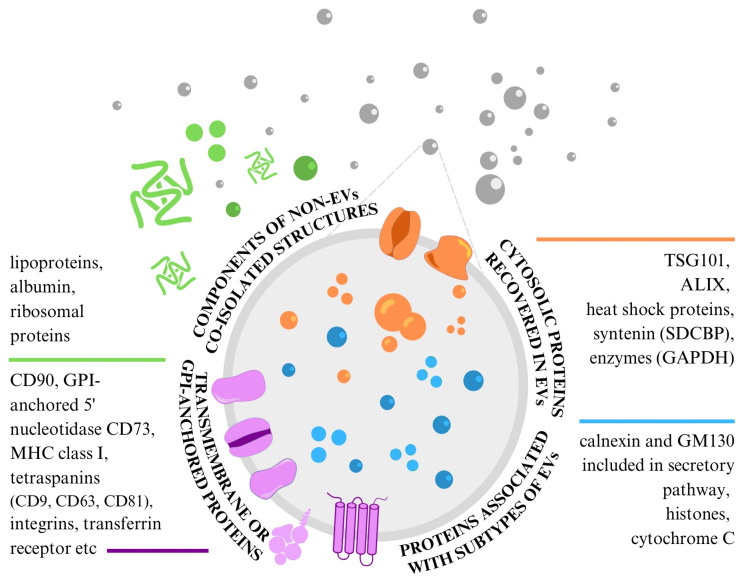
Protein content-based EV characterisation specific to MSC-EVs.

**Table 1 ijms-25-03439-t001:** Summarised recommendations for reporting of EV studies according to updated MISEV2023 guidelines. They suggest quantitative measurements of the EV source, EV number and components, setting out the degree of purity and providing the method’s LoD.

	Updated MISEV2023 Guidelines
Cell culture-conditioned medium	Report medium composition and preparation, characteristics of producing cells, culture conditions and harvesting and storage methods.
EV separation and concentration	Describe used source material, concentration method and conditions; used technology and settings that allow replication; and measurements used to assess separation process.
Western blotting, flow cytometry, MS, ELISA, total protein detection	Define method LoD, describe EV preparation, use controls, report results in normalised units, optimise instrument settings, include antibody information, and provide uncropped images of Western blots.
NTA, DLS, RPS	Report method LoD, instrument settings and all preanalytical procedures; use orthogonal measurements and controls; and report diameter distribution rather than average measurements.
Nucleic acid and lipid detection	Report method LoD, consider co-isolated components, describe EVs’ preparation and treatment before analysis and report sequences of primers and analysis methods. Consider using protein co-localisation.
Morphological studies	Describe the instrument and analysis/acquisition settings, sample preparation process (fixation, adsorption, staining), EV immobilisation method, parameters for the recognition of objects, etc.

## Data Availability

All data are given in the manuscript.

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
