# Peer review of "Current Methods for Analysing Mesenchymal Stem Cell-Derived Extracellular Vesicles"

_ijms, 2024, doi:10.3390/ijms25063439_

Round 1

Reviewer 1 Report

Comments and Suggestions for Authors

This manuscript reviews the analysis methods of EVs. While there is a growing interest in EVs and their therapeutic applications, the lack of standarrd criteria for their purity and effective moiety is a continuous bottleneck of their clinical use. In this sense, this manuscript comprehensively provides an overview of analysis methods for translation/therapeutic applications.

Nonetheless, there are some minor improvements needed before publication. Please consider the following comments:

1) Some of the earlier key protemic stidues on MSC-derived mcirovesicle/exosomes need to be cited.

Extrusion methods may have pros and cons for their applicability to clinical settings and this should be described in more detail. For example, inside-out vesicles generated by extrusion method maybe easy target for phagocytic macrophages.
2) Conclusion with a suggestion regarding a standard criterion for EVs analysis is highly recommended.

Comments on the Quality of English Language

It is needed proof reading from native speaker.

Reviewer 2 Report

Comments and Suggestions for Authors

In this review, the authors offer an insightful overview of various methodologies employed in characterizing MSC-derived EVs, presenting what the reviewer perceives as an innovative perspective. However, a limitation identified is the lack of specificity regarding the unique characteristics of MSC-derived EVs. I would like to recommend a major revision before acceptance.

1. The authors are recommended to include a comprehensive summary of the number (or representative cases) of clinical trials involving MSC-derived EVs conducted in recent 5 years.

2. An updated summary of the "MISEV2023" guidelines is essential.

3. The characterization techniques for EVs often are utilized in combination and reflect a broader range of methodologies to underscore their diverse features. For example, while surface marker analysis is crucial for identifying different EV subpopulations, it is often complemented by particle size characterization techniques such as NTA or Qnano (i.e., 10.1002/advs.202302622). Moreover, the inclusion of morphological studies, employing methods like SEM and cryo- cryo-TEM and AFM, highlights the importance of integrating various characterization tools to provide a comprehensive understanding of EVs' physical and functional properties (i.e., doi.org/10.1002/pmic.201800166). The authors are recommended to introduce more recent works and discuss the significance of introducing synergistic application of these characterization methodologies.

4. The review should elucidate the distinctions in characterization methodologies between MSC-derived EVs and those derived from other cell types. Highlighting any unique attributes of MSC-derived EVs in comparison to other cells-derived EVs would significantly contribute to understanding their specific roles and potential therapeutic applications.

Round 2

Reviewer 2 Report

Comments and Suggestions for Authors

The authors have satisfactorily addressed my concerns.